# Common Principles and Specific Mechanisms of Mitophagy from Yeast to Humans

**DOI:** 10.3390/ijms22094363

**Published:** 2021-04-22

**Authors:** Rajesh Kumar, Andreas S. Reichert

**Affiliations:** Institute of Biochemistry and Molecular Biology I, Medical Faculty and University Hospital Düsseldorf, Heinrich-Heine-University Düsseldorf, 40225 Düsseldorf, Germany

**Keywords:** mitophagy, PINK1, PARKIN, autophagy, quality control, cancer, ubiquitin

## Abstract

Mitochondria are double membrane-bound organelles in eukaryotic cells essential to a variety of cellular functions including energy conversion and ATP production, iron-sulfur biogenesis, lipid and amino acid metabolism, and regulating apoptosis and stress responses. Mitochondrial dysfunction is mechanistically linked to several neurodegenerative diseases, cancer, and ageing. Excessive and dysfunctional/damaged mitochondria are degraded by selective autophagic pathways known as mitophagy. Both budding yeast and mammals use the well-conserved machinery of core autophagy-related genes (*ATGs*) to execute and regulate mitophagy. In mammalian cells, the PINK1-PARKIN mitophagy pathway is a well-studied pathway that senses dysfunctional mitochondria and marks them for degradation in the lysosome. PINK1-PARKIN mediated mitophagy relies on ubiquitin-binding mitophagy adaptors that are non-ATG proteins. Loss-of-function mutations in *PINK1* and *PARKIN* are linked to Parkinson´s disease (PD) in humans, and defective mitophagy is proposed to be a main pathomechanism. Despite the common view that yeast cells lack PINK1- and PARKIN-homologs and that mitophagy in yeast is solely regulated by receptor-mediated mitophagy, some studies suggest that a ubiquitination-dependent mitophagy pathway also exists. Here, we will discuss shared mechanisms between mammals and yeast, how mitophagy in the latter is regulated in a ubiquitin-dependent and -independent manner, and why these pathways are essential for yeast cell survival and fitness under various physiological stress conditions.

## 1. Introduction

Mitochondria are highly dynamic double-membrane surrounded organelles that are essential for eukaryotic life. In mammalian cells, mitochondria generate most of the cellular ATP (∼90%) by oxidative phosphorylation (OXPHOS) [1]. Besides this important bioenergetic function, mitochondria regulate other essential cellular functions such as β-oxidation of fatty acids, heme and phospholipids biosynthesis, amino acid metabolism, redox homeostasis, stress response, and cell fate decisions [2]. The spatial organization and function of mitochondria rely on both nuclear- and mitochondria-encoded proteins, which act together, very often as multimeric protein complexes, to fulfill a pleiotropy of mitochondrial functions [3]. Mitochondria are very diverse in structure and highly dynamic both at an intracellular as well as at an intramitochondrial level [4].

Errors in the spatial organization of mitochondria can result in mitochondrial dysfunction, which is deleterious for the life of cells and organisms. For instance, defects in respiratory chain complexes promote the generation of reactive oxygen species (ROS) and the loss of the membrane potential (Δ*ψ*) across the inner mitochondrial membrane. Mitochondria can also produce ROS as by-products of aerobic OXPHOS [2]. Mitochondrial ROS can damage mitochondrial proteome and lipids and even cause mitochondrial DNA (mtDNA) mutations. The dissipation of the mitochondrial membrane potential also compromises mitochondrial function owing to defects in mitochondrial protein import.

Cells have evolved several mitochondrial quality control mechanisms to restore and preserve the fitness of mitochondria in responses to varying degrees of mitochondrial damage [5,6,7,8,9]. Low or transient mitochondrial damage can be repaired by either intraorganellar protein quality control machineries or machineries, which work at the organelle’s surface in the cytosol. However, when damage is prolonged or severe, mitochondria are selectively recognized and degraded via a process known as mitophagy [9]. In this article, we will discuss the emerging roles of mitophagy under different physiological or stress conditions. We will highlight the multiple levels of mitophagy regulation in yeast and human, focusing on the common and general principles as well as the specific aspects in this regard.

## 2. Mitochondrial Quality Control at Multiple Levels

Mitochondrial function declines with age and is a pathological hallmark of several neurodegenerative disorders, diabetes, and cancer [10]. For instance, neuronal cells rely heavily on proper mitochondrial function owing to the high requirement for Ca^2+^ buffering and ATP production at the synapse [6]. One mechanism suggested to contribute to mitochondrial dysfunction is impaired mitochondrial quality control [11,12]. Hence, it is not surprising that mitochondrial function is compromised during aging. However, different surveillance mechanisms have evolved in response to cellular stress to maintain mitochondrial fitness at the molecular, organellar, and cellular levels [6,7]. Eukaryotic cells have four main mitochondrial quality control mechanisms that function via intraorganellar proteostasis (the mitochondrial protease–chaperone network), the cytosolic ubiquitin-proteasome system (UPS), mitochondrial-derived vesicles (MDVs)/lysosome, and lastly mitophagy in response to low-to-high levels of mitochondrial damage [6,7] (Figure 1).

The mitochondrial protease–chaperone network serves as the first line of defense at the molecular level that removes misfolded or damaged proteins in mitochondria when mitochondrial damage is low [6,7]. In addition, *S. cerevisiae* mitochondria can serve as essential proteolytic compartments to import and degrade cytosolic misfolded protein aggregates by the matrix localized ATP-dependent Lon proteases [13]. Human mitochondrial DNA (mtDNA) encodes 13 subunits (∼1% of mitochondrial proteome) of four OXPHOS complexes (complexes I, III, IV, and V) [2]. Thus, 99% of mitochondrial proteins are encoded by the nuclear genome, synthesized in the cytosol as precursor proteins, and imported into mitochondria by different import pathways [2]. However, when the capacity of mitochondrial protein import machineries is overwhelmed in cells, possibly at high energy-demand conditions or when mitochondria are energetically compromised, this leads to the accumulation of nuclear-encoded mitochondrial precursor proteins at the organelle´s surface [14,15]. Cells evolve several protein quality control pathways that remove stalled mitochondrial precursor proteins by SUMO/Ubiquitin-mediated proteasomal degradation and prevent their misfolding/aggregation on the organelle’s surface [14,15,16]. These surveillance pathways protect mitochondrial functions during import stress.

In response to mild/local mitochondrial damage, mitochondria also release small vesicles (∼100 nm diameter) known as mitochondrial-derived vesicles (MDVs) or mitochondrial-derived compartments (MDCs), which incorporate aberrant mitochondrial components [7,8]. MDVs can communicate with lysosomes/peroxisomes, while MDCs facilitate organelle homeostasis by removing outer mitochondrial membrane (OMM) and/or inner mitochondrial membrane (IMM) misfolded/damaged proteins from mitochondrial membranes [7,8]. Thus, lysosome-targeted MDVs remove damaged mitochondrial portions similar to MDCs and represent a third mitochondrial quality control mechanism, working at the organellar level [7,8,17].

When these quality control mechanisms are insufficient, e.g., owing to severe mitochondrial damage or depolarization at the inner membrane, individual dysfunctional organelles are segregated from the healthy network and subsequently selectively degraded through mitophagy [8,18]. It is important to note that the selective spatial isolation of mitochondria by mitochondrial fission is initially ensured by a rapid inactivation of mitochondrial fusion by stress-induced proteolytic processing of the fusion factor OPA1 [19,20]. Thus, alteration of mitochondrial morphology and subsequent mitophagy is perceived as a cytoprotective mechanism to reduce intracellular mitochondrial dysfunction and to prevent cell death/apoptosis at the cellular level.

### 2.1. Mitophagy—An Overview

The first report showing the presence of mitochondria within autophagosomes in mammalian cells was demonstrated in 1957 [21]. The term “mitophagy” for the selective turnover of damaged and/or superfluous mitochondria by autophagy machinery was coined by John Lemasters [22]. Mitophagy is crucial for maintaining mitochondrial quality control and limiting somatic mitochondrial DNA (mtDNA) mutations with aging. Indeed, mitophagy is a fascinating pathway as it is directly linked to cellular metabolism, differentiation, physiology, and a broad spectrum of pathologies. For instance, during red blood cell (RBC) maturation, mitochondria are removed by Nix-dependent mitophagy, where Nix (also called Bnip3L) acts as a mitophagy receptor [23,24]. Moreover, Nix-/- mice accumulate mitochondria and developed mild anemia with reduced mature RBCs from erythroid precursor cells [23,24]. PARKIN, an E3 ubiquitin ligase encoded by the *PARK2* gene, is implicated with Parkinson’s disease (PD). Several loss-of-function mutations in the *PARK2* gene were detected in PD patients. PARKIN is recruited selectively to damaged mitochondria and promotes mitophagy by ubiquitination of its substrates at the outer mitochondrial membrane (OMM). Thus, mitophagy has an important role in development, health, and disease. Over the last decade, compelling evidence from yeast and mammalian cells showed that the removal of damaged/superfluous mitochondria from cells is specific [9]. Several mitophagy regulatory factors are recruited on the outer surface of mitochondria and promote their recognition and sequestration into autophagosomes for clearance. Mitophagy can prevent the accumulation of dysfunctional/damaged mitochondria within cytosol, and thus limits an increase of ROS levels or pro-apoptotic factors.

In response to stress and starvation, eukaryotic cells often elicit an evolutionarily conserved non-selective autophagy to degrade and recycle cytosolic constituents [25]. The cargo for autophagy (organelles and macromolecules) are randomly sequestered into double-membrane autophagosomes. The autophagosome’s outer membrane subsequently fuses with lysosomes (or vacuoles in yeasts) for substrate degradation [25]. While starvation-induced autophagy is an early response (after ∼2 h of starvation), mitochondria are turned over selectively at later stages of starvation (e.g., ∼12–24 h), employing the core autophagy complex, Atg1-Atg13-Atg17-Atg31-Atg29 [26,27,28]. However, it is not fully understood why mitochondrial degradation occurs delayed to bulk autophagy during prolonged starvation in yeast and mammals. One possibility is that substrate specificity may partly be determined by the steric hindrance of the substrates where smaller substrates are degraded rapidly, while larger substrates are turned over at later time points during amino acid starvation. Why and how autophagosomes sense, select, and sequester specific substrates in an ordered fashion upon starvation is still unclear.

### 2.2. Mitophagy in Yeast

In baker’s yeast (*Saccharomyces cerevisiae*), mitophagy is induced by prolonged respiratory growth or a shift from respiration to nitrogen starvation [26,29]. In addition, yeast mitophagy can be induced in respiring cells by treatment with rapamycin, an inhibitor of the mammalian target of rapamycin (mTOR) [30]. Respiration is believed to be a prerequisite for mitophagy under these conditions, but the reason for this is unclear. It could be linked to increased oxidative stress, which was reported to increases steady-state levels of Atg32, a mitochondria-anchored receptor required for mitophagy. Interestingly, N-acetylcysteine (NAC), an antioxidant treatment, reduced Atg32 levels by ∼2–3-fold, significantly suppressing mitophagy [29]. During respiratory growth, Atg32 is maximally induced within the mid-log phase (30 h), then reduced in the post-log phase (36–72 h) [29]. Thus, Atg32 is temporally upregulated during respiratory growth conditions and subsequently degraded in an autophagy-dependent and -independent fashion [29]. However, it raises the question of whether other mitophagy regulators are expressed or activated to promote mitophagy during respiratory conditions when Atg32 is almost depleted in cells (e.g., 72 h of growth) [29].

Yeast mitophagy depends on the adaptor protein Atg11 and the receptor protein Atg32 [26,29] (Figure 2). Atg32 is specifically involved only in mitophagy and does not regulate other autophagy types such as bulk autophagy or the cytoplasm-to-vacuole targeting (CVT) pathway under mitophagy-inducing conditions. However, the *atg32*∆ mutant did not show any detectable growth phenotypes under mitophagy-inducing conditions, raising the question of whether Atg32-dependent mitophagy is physiologically essential for yeast cell survival and stress resistance [26].

Atg32 is an integral membrane protein that exposes its N-terminal domain towards the cytosol and the C-terminal domain in the mitochondrial inter-membrane space (IMS) [29].

Interestingly, the cytosolic N-terminal domain of Atg32 has a WXXI/L/V, a conserved Atg8/LC3-interacting motif (AIM/LIR, residues WQAI^86–89^), that anchors mitochondria to Atg8 at the phagophore assembly site (PAS) [29]. Importantly, an Atg32^AQAA^ mutant where Trp-86 and Ile-89 each were mutated to Ala showed no binding to Atg8 in yeast-two-hybrid assays [29]. The Atg32^AQAA^ mutant showed only partial defects in the kinetics of mitophagy (∼88% of the wild-type level) [29]. However, the LIR docking site (LDS) mutant of Atg8 (i.e., Atg8^P52A/R67A^) significantly reduces Atg32-dependent mitophagy (∼60% of the wild-type level) when combined in trans with the Atg32^AQAA^ mutant. This observation raises the possibility that Atg32 may have additional functional AIM/LIR motif(s), which can provide a high-avidity interaction of Atg32 to the Atg8-coated phagophore surface during mitophagy. Indeed, three AIM/LIR motifs (the canonical AIM1, WEEL^412–415^ and two cryptic upstream AIM2, FYSF^376−379^ and AIM3, LPEL^384−387^) were found in Atg19, a well-studied selective autophagy receptor of the CVT pathway, increasing the avidity for its interaction with Atg8 at the PAS [31]. In addition, Atg32-Atg8 interaction in *S. cerevisiae* might also be regulated via phosphorylation of Atg32 at potential phosphorylation sites (S81/83/85) upstream of the AIM/LIR motif (WQAI^86–89^). Indeed, in *P. pastoris*, such Atg32-Atg8 interaction is regulated by phosphorylation at Thr^119^ that is present upstream of the Atg32 AIM motif (WQMV^121−124^) [32].

Atg32 has no apparent mammalian homolog based on amino acid sequence similarity. An Atg32 functional homolog was proposed to exist in mammals with the following molecular features: mitochondrial localization; WXXL/I/V motifs; LC3 interaction; clusters of acidic amino acids (D/E); and single membrane-spanning topology [29]. Using these molecular profiles of Atg32, recently, Bcl2-like 13 (Bcl-2-L-13) was identified as a functional homolog of Atg32 by screening with UniProt database (http://www.uniprot.org/) as a search tool. Mouse Bcl-2-L-13 is a 434 amino acids protein that contains a C-terminal single transmembrane domain (TMD) and one functional LC3-interacting region (LIR) at residues 273–276 (WQQI) at its N-terminal domain that faces cytosol [33] (Figure 2). Upon mitochondrial damage by carbonyl cyanide m-chlorophenylhydrazone (CCCP), Bcl-2-L-13 is localized to mitochondria and promotes mitophagy, independent of the E3 ubiquitin ligase PARKIN. Thus, Bcl-2-L-13 can function as a mammalian mitophagy receptor and partially rescues mitophagy defect when exogenously expressed in *atg32*Δ yeast [33]. Whether Bcl-2-L-13-mediated mitophagy has a major physiological significance in humans or mice is still unclear.

### 2.3. Phosphorylation Regulates Atg32 Activity

The cytosolic N-terminal domain of Atg32 acts as a “degron” signal as it harbors four essential features for mitophagy: first, an AIM/LIR motif (WQAI^86–89^) [29]; second, Atg11-binding motif (aa 51–150 including phosphorylated S114 and S119) near the AIM/LIR [34]; third, a binding region (aa 151–200) for PP2A (protein phosphatase 2A)-like protein phosphatase Ppg1, a negative regulator of Atg32 phosphorylation (i.e., mitophagy) [35]; and the fourth region (aa 200–341), called the pseudo-receiver (PsR) domain, is required for mitophagy by an unknown mechanism [36] (Figure 2). Thus, Atg32 binds to Atg8 (LC3 in mammals), conjugated to phosphatidylethanolamine (PE) at the PAS. The lipid-conjugated Atg8 (Atg8-PE) is an essential rate-limiting factor for nucleation and elongation of the PAS during autophagosome formation.

Atg32 physically interacts with Atg8 and Atg11, an adaptor protein to confer selectivity for mitophagy [26,29,37]. Recent studies showed that Atg32-mediated mitophagy is controlled by reversible and dynamic protein phosphorylation. Casein kinase 2 (CK2) directly phosphorylates Atg32 at two residues, Ser114 and Ser119, within the Atg11-binding motif and activates mitophagy in yeast [34,38]. CK2-mediated Atg32 phosphorylation stabilizes the Atg32-Atg11 interaction that subsequently initiates core Atg protein assembly and autophagosome formation [38]. Interestingly, CK2 does not control bulk autophagy, the CVT pathway, or pexophagy, suggesting it is one of the specific mitophagy regulators [38]. In contrast, the protein phosphatase 2A (PP2A)-like protein phosphatase Ppg1 transiently binds to Atg32 (amino acid residues 151–200) and dephosphorylates Ser114 and Ser119, leading to compromised mitophagy [35]. Deletion of the Ppg1 binding region (aa 151–200) in Atg32 shows the same phenotypes as *ppg1*Δ, resulting in constitutive interaction of Atg32 with Atg11 and enhanced mitophagy by 3–4-fold compared with wild-type cells [35]. Interestingly, Ppg1 specifically regulates mitophagy without affecting bulk autophagy, the cytoplasm to vacuole targeting (CVT) pathway, or pexophagy [35]. Thus, the Ppg1 protein phosphatase suppresses excessive mitophagy under normal growth conditions by counteracting CK2-mediated phosphorylation of Atg32. However, under mitophagy-inducing conditions, Ppg1 phosphatase activity is suppressed, allowing CK2 to phosphorylate Atg32 [35]. Thus, Ppg1 emerges as a novel central regulator for Atg32-dependent mitophagy; it raises some key questions for future research directions. For instance, what are the underlying molecular mechanisms by which Ppg1 phosphatase activity is suppressed, allowing CK2 (a constitutively active protein kinase) for Atg32 phosphorylation under mitophagy-inducing conditions.

In addition to protein kinase CK2, two mitogen-activated protein kinase (MAPK) signaling cascades, Wsc1–Pkc1–Bck1–Mkk1/2–Slt2 and Ssk1–Pbs2–Hog1, are required for mitophagy in yeast [34,39]. Hog1, but not Slt2 kinase, functions upstream of CK2 to regulate Atg32 phosphorylation and mitophagy flux [34]. However, the underlying molecular mechanism by which these two MAPK-signaling pathways regulate CK2-mediated phosphorylation of Atg32 is still unclear.

Atg32 is modified at both N and C-terminus after its recruitment at the mitochondria. Upon mitophagy induction, the mitochondria-localized Atg32 is phosphorylated by CK2 at its cytosolic N-terminus, which is essential to form a complex with Atg8 and Atg11 at the PAS. In addition, Atg32 is proteolytically cleaved at its C-terminus by the inner mitochondrial membrane i-AAA (intermembrane space ATPases associated with diverse cellular activities) protease, Yme1 [40]. This processing is compromised when Atg32 is either tagged at its C-terminus in wild-type (WT) cells, or Yme1 is deleted/mutated to its catalytic inactive form with a single point mutation E541Q [40]. This causes defective mitophagy caused by nitrogen starvation. The exact mechanism of Yme1-mediated mitophagy is not clear. However, in *yme1*Δ cells, the interaction between Atg32 and Atg11 adaptor is reduced, leading to impaired recruitment of mitochondria to the PAS. However, other studies indicate no mitophagy defect in cells lacking Yme1, suggesting that Yme1-dependent processing may be strain and/or condition-specific [41,42]. Nevertheless, these two processes, CK2-mediated N-terminal phosphorylation and Yme1-dependent C-terminal processing of Atg32, control nitrogen starvation-induced mitophagy.

The function of Yme1 under post-log/stationary phase mitophagy is different from that under nitrogen starvation. Yeast cells that are grown for 2–3 days (post-log/stationary phase) in non-fermentable carbon sources (respiratory media) such as lactate/glycerol/ethanol tend to proliferate and produce reactive oxygen species (ROS). Therefore, cells promote selective mitochondrial degradation to limit the abundance of mitochondria (healthy or damaged). Cells that lack Yme1 showed severe mitochondrial damage (defective morphology) revealed by transmission electron microscopy (TEM) associated with the vacuolar rim [43]. This raises the question of whether damaged mitochondria in *yme1*Δ cells are degraded by microautophagy or still depend on Atg32-mediated macromitophagy. However, *yme1*Δ cells showed a ~1.5–3.0-fold greater mitophagy rate than wild type (WT) cells grown on a non-fermentable the carbon source for 2–3 days [41,43].

### 2.4. Mitochondrial Fission and Yeast Mitophagy

As intact mitochondria, present as interconnected tubules in many living cells, have larger dimensions than autophagosomes, sequestration of damaged mitochondria may be facilitated subsequent to mitochondria fragmentation. It is shown that the mitochondrial fission machinery separates damaged/dysfunctional mitochondria from the healthy mitochondrial network in mammalian cells [19,20,44,45]. Such a spatial separation promotes mitophagy by selective sequestering of smaller damaged mitochondria within autophagosomes and subsequent lysosomal degradation [44,45]. Thus, blocking of mitochondrial fission leads to defective mitophagy and accumulation of damaged mitochondria in mammalian systems [44]. Is mitochondria fission machinery a prerequisite for mitophagy in yeast? In *S. cerevisiae*, mitochondrial fission is mediated by the fission factors Dnm1 and Fis1 [46]. Cells lacking either DNM1 or FIS1 significantly reduce nitrogen starvation-induced mitophagy [46]. This is consistent with the previous study where deletion of DNM1 significantly suppresses mitophagy in mdm38∆ cells [47]. Still, rapamycin-induced mitophagy was shown not to depend on Dnm1 or Fis1 [48]. Atg11 was reported to recruit Dnm1 to “marked” mitochondria that are destined for degradation [46]. It was also suggested that the ER-mitochondria encounter structure (ERMES) complex might also participate in mitochondrial fission during mitophagy [46,49]. However, the exact molecular mechanism by which mitochondrial fission machinery and as yet unidentified fission factors can regulate the early step of mitophagy in yeast is still unclear. Thus, in an updated mitophagy model, Atg32 recruits Atg11 to “marked” mitochondria upon induction of mitophagy. Atg11 subsequently interacts and recruits Dnm1 and other fission components to these “marked” mitochondria and facilitates their fragmentation [46]. These fragmented mitochondria are then transported to the PAS, where other core Atg proteins are recruited, initiating the autophagosome formation. In mammalian cells, upon energy stress, MFF (mitochondrial fission factor) recruits Drp1 (dynamin-related protein 1), a GTPase (lacks hydrophobic transmembrane domain), from the cytosol to the mitochondrial outer membrane and catalyzes mitochondrial fragmentation for efficient autophagosomal engulfment of mitochondria and mitophagy [50]. Notably, the Drp1-MFF interaction and mitochondrial fission are mediated after adenosine monophosphate (AMP)–activated protein kinase (AMPK)-mediated phosphorylation of MFF at Ser^155^ and Ser^172^ [50].

### 2.5. Transcriptional and Translational Regulation of Atg32 Activity

Upon mitophagy induction, Atg32 transcripts and protein levels are controlled by transcriptional and co-translational regulation. It has recently been shown that Ume6–Sin3–Rpd3, a transcriptional repressor complex, directly binds to the promoter (URS1 consensus 5′-TCGGCGGCT-3′) of ATG32 and ATG8 ∼2.5 times higher than negative control TFC1 promoter (Figure 2). The yeast cells lacking either SIN3 or RPD3 or UME6 showed a ∼2.5-fold higher expression of ATG32 and ATG8 mRNAs. Thus, Ume6–Sin3–Rpd3 complex negatively regulates autophagy and mitophagy. In addition, co-translational N-terminal protein acetylation (Nt-acetylation) also regulates mitophagy under respiratory growth conditions [51].

Co-translational Nt-acetylation is a widespread irreversible modification of proteins in eukaryotes, affecting 50–70% of the yeast proteome and approximately 90% in higher eukaryotes. Nt-acetylation is achieved as soon as nascent polypeptide chains emerge at the ribosome exit tunnel during translation. Nt-acetylation is accomplished by the major cytosolic N-terminal acetyltransferase A (NatA) complex, which binds to the large ribosomal subunit near the ribosome exit tunnel. Yeast NatA complex is comprised of one catalytic subunit, Ard1, and the other adaptor subunit, Nat1. The NatA complex catalyzes acetylation of the second amino acids, Ala, Val, Ser, Thr, Gly, and Cys, of the nascent polypeptide once the N-terminal methionine residues are co-translationally cleaved by methionine-amino peptidase (MetAP). Nt-acetylation targets nascent proteins for degradation/stability, folding, protein–protein interactions, and translocation to subcellular compartments.

Loss of Nt-acetylation in natAΔ yeast cells by deleting either Ard1 or Nat1 or both proteins showed a drastic reduction (more than 90%) in mitophagy without affecting bulk autophagy. These mutants mimic a defective mitophagy phenotype of the *atg32*∆ mutant. How does the NatA complex regulate mitophagy in yeast? What are the endogenous substrates of NatA that control mitophagy without affecting core autophagy machinery? Interestingly, NatA is estimated to N-terminally acetylate ∼40% of the yeast and mammalian proteome. Atg32 seems to be a direct substrate for NatA-mediated Nt-acetylation because it contains “Val” at the second amino acid position. However, the substitution of valine to proline (V2P) in Atg32 can prevent Nt-acetylation by NatA, which was still functional in promoting mitophagy as wild-type Atg32. This suggests that Atg32 is not a direct substrate of NatA for Nt-acetylation during mitophagy. Therefore, an indirect regulation was proposed for NatA-mediated Nt-acetylation during mitophagy. NatA may acetylate Rpd3 and Sin3 transcription repressor complex on valine and serine at their second position, respectively. Nt-acetylation of Rpd3 and Sin3 can serve as a specific degradation signal for polyubiquitination and proteasomal degradation via the Ac/N-end rule pathway. Thus, it is plausible that NatA may reduce Rpd3 and Sin3 transcription repressors’ protein half-life to enhance transcription of ATG32 and ATG8 during mitophagy. This study warrants further research to understand how Nt-acetylation can regulate mitophagy by acetylating yet-undiscovered substrates among ∼40% of the yeast proteome. Future studies can dissect what fraction of mitochondrial, cytosolic, or nuclear proteins is Nt-acetylated by NatA. The Nt-acetylation field is just emerging to understand the molecular mechanism and the physiological relevance of this process during mitophagy and the other environmental stress response pathways.

Furthermore, Atg33 was identified as the second candidate from a genome-wide screen that peculiarly regulates the post-log phase mitophagy [52]. Atg33 may promote degradation of aged mitochondria; however, the exact function of Atg33 in mitophagy is still unclear.

### 2.6. Yeast Model of Mitochondrial Damage and Mitophagy

In yeast, bona fide models that can mimic mitochondrial damage and initiate mitophagy are still lacking. It appears that yeast is resistant to mitophagy induction by applying known mitochondrial toxins that impair OXPHOS [48]. However, some studies could show that mitophagy can be induced by mitochondrial damage in this model organism by genetic approaches [47,53,54].

The sphingolipid metabolism pathway is conserved from yeast to human, where a small amount of bioactive lipid ceramide is locally synthesized at the cytosolic side of the endoplasmic reticulum (ER) and mitochondria [55]. Several enzymes of sphingolipid metabolism, including ceramide synthase and reverse ceramidase, were shown to be localized to ER-mitochondria contact sites [55]. Interestingly, yeast and mammalian cells synthesize ceramides from sphingolipids under environmental stress conditions such as heat shock [56] and serum starvation [57].

Ceramides regulate mitochondrial function and morphology/dynamics via direct interaction with the electron-transport chain (ETC) [58]. In addition, mitochondrially localized ceramides were shown to regulate mitochondrial translation at least for a few subunits such as COX3 [58]. *Saccharomyces cerevisiae* Isc1, an inositol phosphosphingolipid phospholipase C involved in de novo ceramides and phytoceramides biosynthesis from the complex sphingolipids. Isc1 translocates to the mitochondria when respiration is induced by a transition from fermentation to nonfermentable carbon sources (e.g., glycerol or lactate) [58]. Yeast cells lacking Isc1 show mitochondrial fragmentation and reduced chronological life span (CLS) owing to failed mitochondrial translation and perturbation in sphingolipid metabolism [58,59].

Yeast *isc1*Δ cells reduce α-hydroxylated phytoceramides’ synthesis by ∼90% from mitochondria and show hypersensitivity towards oxidative stress (H_2_O_2_) and ethidium bromide (EtBr) [60]. The *isc1*Δ mutant consequently causes respiratory deficient, “petite” phenotype [60]. Recently, it was shown that yeast *isc1*Δ cells, when grown at post-log phase (respiratory condition), led to compromised mitochondrial function and fragmentation by the fission factor Dnm1 (Drp1 in mammals) [59]. Under these conditions, Dnm1 is also induced, suggesting Dnm1 is essential to fragment the damaged mitochondria [59]. Mitochondrial fragmentation eventually hyperactivates mitophagy (Atg32-dependent) as an adaptive stress response. Furthermore, ATG32 deletion in an *isc1*Δ background (i.e., *isc1*Δ*atg32*Δ mutant) further exacerbates the growth defect and results in ∼40% CLS reduction of *isc1*Δ cells [59]. Under the identical condition, the mitophagy-deficient mutant atg32Δ essentially behaves as the wild-type strain without showing any detectable growth phenotype. These observations suggest that Atg32 is physiologically essential to promote degradation of damaged mitochondria and maintain cell growth and survival of ceramide-deficient *isc1*Δ mutant (Figure 3). Thus, Atg32-dependent mitophagy prevents excessive cell death or premature ageing caused by severe organelle damage in an *isc1*∆ mutant [59]. Yeast *isc1*∆ cells in response to mitochondria damage also activate the mitogen-activated protein kinase (MAPK) Hog1 (homologue of mammalian p38) required for mitophagy [39,59]. Phosphorylation of mitophagy receptor Atg32 at Ser-114/119 by CK2 is crucial for Atg32–Atg11 interaction and mitophagy [38]. Hog1 is required for Atg32 phosphorylation, but it cannot directly phosphorylate Atg32, suggesting that Hog1 functions upstream of CK2 [39]. However, the molecular mechanism by which Hog1 can control CK2-mediated Atg32 phosphorylation is still unclear.

How do yeast *isc1*Δ cells enhance mitophagy mechanistically compared with wild-type (WT) cells? In HL-60 human cells, it was reported that ceramides directly activate mitochondrial protein phosphatase 2A (PP2A), which can dephosphorylate its substrate, Bcl2 [61]. Bcl2 is an anti-apoptotic protein whose function relies on phosphorylation at an evolutionarily conserved site, serine 70 [61]. Therefore, high ceramide concentration can lead to apoptotic cell death owing to the dephosphorylation of Bcl2 by PP2A. In contrast, Kanki´s group recently showed that Atg32 phosphorylation is regulated by PP2A-like phosphatase, Ppg1 [35]. Ppg1 functions as a negative regulator of Atg32 phosphorylation as it dephosphorylates Atg32 and inhibits mitophagy without affecting other selective autophagy pathways [35]. Therefore, it is plausible to hypothesize that *isc1*Δ cells may inactivate Ppg1 because of reduced endogenous ceramide levels, leading to the hyperactivation of mitophagy. However, it is not entirely understood if Ppg1 localizes near its substrate Atg32 or mitochondria to control mitophagy.

### 2.7. Ubiquitin-Dependent Regulation of Mitophagy

Eukaryotic cells have several mitophagy mechanisms, regulated via multiple signaling cascades under different mitochondrial and cellular stress conditions. Mitophagy pathways are classified as ubiquitin-dependent and -independent. Most of the core autophagy-related proteins that are involved in mitophagy are conserved from yeast to humans. The well-characterized PINK1–PARKIN pathway regulates ubiquitin-dependent mitophagy, which is conserved in metazoans, such as *Drosophila melanogaster*, *Caenorhabditis elegans*, and mammals [37,62]. There are no apparent PINK1 or PARKIN homologs in bacteria, yeast, or plants.

### 2.8. The PINK1–PARKIN Pathway in Mammals

In many metazoan cell types, mitophagy is regulated by PTEN-induced putative kinase protein 1 (PINK1) and PARKIN, a RING-HECT hybrid E3 ubiquitin ligase (which have no yeast homolog), and loss-of-function mutations in two genes encoding these proteins are linked to Parkinson’s disease (PD), the second most common neurodegenerative disease in humans. Together, these two proteins safeguard a protective mitophagy response to mitochondrial stress and limit the accumulation of damaged/toxic mitochondria. In cells bearing healthy mitochondria, PINK1 (a Ser/Thr protein kinase; also known as PARK6) is rapidly degraded via the N-end rule pathway after partial import into mitochondria, PARL-dependent processing and retranslocation [63]. PARKIN (PARK2, also known as PRKN) remains in an autoinhibited state in the cytosol [5,64,65,66,67]. Thus, PINK1 is barely detectable in cells with healthy mitochondria. However, upon mitochondrial damage (e.g., mitochondrial depolarization), active PINK1 is stabilized on the mitochondrial surface and recruits the E3 ubiquitin ligase PARKIN [68,69]. Subsequently, PINK1 phosphorylates its substrates, ubiquitin (Ub), at the conserved residue Ser65 (generating pSer65-Ub) and the residue Ser65 in the N-terminal Ub-like (UBL) domain of PARKIN (generating pSer65-PARKIN) [70,71,72,73]. Thus, phosphorylated ubiquitin pSer65-Ub is reversible and barely detectable in basal conditions, but rapidly induced by mitochondrial damage in cells and amplified by functional PARKIN [74].

Interestingly, when pSer65-Ub is associated with pSer65-PARKIN, it enhances PARKIN´s ubiquitin ligase activity (∼4400-fold) by retaining PARKIN on the MOM, which can provide additional ubiquitin molecules for PINK1 phosphorylation, creating the feedforward loop [5,75] (Figure 4). Active PARKIN then ubiquitinates its substrates, local mitochondrial outer membrane (MOM) proteins, which can recruit specific ubiquitin-binding mitophagy receptors [5]. Ubiquitin-binding mitophagy receptors recruit ATG8 (LC3)-positive autophagosomes, allowing selective sequestration of damaged mitochondria and subsequent lysosomal degradation.

PARKIN generates K6 (preferentially), K27, K48, and K63-linked Ub chains on its substrates upon mitochondrial stress [76]. Deubiquitinating enzymes (DUBs) reverse ubiquitin modifications and thus essential negative regulators to ensure that damage signals are strong enough not accidentally degrade healthy mitochondria [77]. The mitochondrially localized DUBs, including USP15, USP30, and USP35, counteract PARKIN-dependent ubiquitylation of selected MOM substrates, and thus limit mitophagy [9,77,78,79,80,81,82,83,84]. In contrast to these DUBs, USP8 preferentially hydrolyzes K6-linked Ub chains from PARKIN and promotes its recruitment to depolarized mitochondria; thus, USP8 acts as a positive regulator of mitophagy [78].

Interestingly, USP30 shows a preference for K6 Ub linkages once it is localized to mitochondria [82]. Knockdown of USP15 or USP30 rescues the mitophagy defect with pathogenic PARKIN mutations in PD patient fibroblasts and *Drosophila*, improving mitochondrial integrity and organismal survival [79,81]. Thus, genetic and pharmacological inhibition of USP15 or USP30 may represent a therapeutic strategy for PD pathology caused by reduced PARKIN levels and defective mitophagy [85].

Does PINK1-dependent ubiquitin phosphorylation at Ser65 impact DUB activity and selectivity? Wauer and colleagues showed that ∼12 DUBs, including USP2, USP8, USP15, USP30, Ataxin-3, and USP21, hydrolyze phosphoUb chains with significantly less activity from in vitro reconstitution assay [86]. In the case of USP30, structural and biochemical analysis shows that phosphorylation of the distal ubiquitin in the K6-linked ubiquitin dimer can preclude access to USP30 [77]. In addition, a single phosphorylation of the distal ubiquitin of a tetraubiquitin chain is sufficient to prevent DUB-mediated hydrolysis [77]. Thus, pSer65-Ub has an additional function beyond PARKIN activation. At mitochondria, pSer65-Ub can make a DUB-resistant mitophagy signal by phosphocapping of K6 Ub chains, preserving recruitment sites for ubiquitin-binding mitophagy receptors that link the mitochondria to autophagosomes. Interestingly, in addition to these DUBs, phosphatase and tensin homolog (PTEN)-long (PTEN-L) was recently identified as a novel negative regulator of the PINK1–PARKIN mitophagy pathway that dephosphorylates pSer65-Ub in vivo and in vitro via its protein phosphatase activity [87] (Figure 4).

In mammalian cells, ubiquitin-binding mitophagy receptors contain ubiquitin-binding domains (UBDs) and a four-residue short hydrophobic sequence, known as LC3-interacting region (LIR) motif ( [W/Y/F]XX [L/I/V]) [5]. These mitophagy receptors recognize and bind to ubiquitylated MOM proteins via their UBD domains on the one hand and to LC3II-conjugated autophagosomal double-membrane via their LIR motifs on the other hand [5]. In mammals, five ubiquitin-binding mitophagy receptors were identified that are linked to mitophagy: p62 (SQSTM1), NBR1, NDP52 (CALCOCO2), optineurin (OPTN), and TAX1BP1 [5,88]. Thus, cells lacking these five receptors (termed Penta KO) fail to remove mitochondria after activating the PINK1–PARKIN pathway [88].

p62 (SQSTM1), the best-characterized and the first identified autophagy cargo receptor, has both a ubiquitin-associated (UBA) domain to interact with ubiquitinated protein substrates and an LIR motif for binding to LC3/GABARAP-positive autophagosomes [89]. Several studies showed that p62 is dispensable for mitophagy initiation without promoting autophagosome biogenesis around damaged mitochondria. Recently, it has been reported that p62 can promote PINK1–PARKIN independent mitophagy [90] by executing juxtanuclear clustering of damaged mitochondria that resemble p62-mediated ‘aggresomes’ of ubiquitinated aggregated proteins [91]. However, various pathogenic PARKIN mutations interfered with p62-mediated clustering of damaged mitochondria, suggesting p62 collaborates with PARKIN for mitochondria clustering at perinuclear regions [91]. Thus, the role of p62 in mitophagy is controversial and requires further study for understanding the functional redundancy of p62 in mitochondrial elimination in a context-specific (e.g., tissue and cell-types) manner. Nevertheless, upregulation of p62 may serve as an attractive therapeutic target to counteract aging and Parkinson’s disease (PD) to restore/enhance alternative mitophagy pathways in pathological conditions where the PINK1–PARKIN pathway is perturbed.

Interestingly, it has been recently reported that OPTN is sufficient to rescue mitophagy in Penta KO cells when OPTN is stably re-expressed in these cells along with mitochondrial-targeted non-cleavable linear lysine-less di-ubiquitin (i.e., mito-2Ub K0) [92]. Mito-2Ub K0 induces mitophagy without any chemical-induced mitochondrial depolarization that is a prerequisite for PINK1-mediated ubiquitin phosphorylation [92]. This study suggests that mito-2Ub K0 can bypass the PINK1–PARKIN pathway and promote mitophagy via preferentially binding with the ubiquitin-associated (UBA) domain of OPTN. Importantly, in addition to binding to ubiquitin and LC3, OPTN also binds with ATG9A vesicles that supply lipids for de novo synthesis of autophagosomal membranes in close proximity to the ubiquitinated mitochondria [92]. The OPTN–ATG9A interaction is mediated by the leucine zipper domain (residues 143–164 aa) of OPTN [92]. Thus, the ubiquitin-OPTN–ATG9A axis functions in concert with the known ubiquitin-OPTN–ATG8 axis in mitochondrial clearance using specific interaction domains within OPTN [92]. In contrast, another study reported that the NDP52 receptor initiates mitophagy by recruiting FIP200, a core component of the autophagy initiation complex (i.e., ubiquitin-NDP52-FIP200 axis) [93]. Thus, among all five mitophagy receptors, only OPTN and NDP52 are crucial for mitochondrial clearance as both recruit the core components of the autophagy machinery to initiate autophagosome biogenesis directly near ubiquitinated cargo [92,93].

Interestingly, it has been shown that p38 (MAPK14), a yeast homolog Hog1, is involved in promoting starvation- or hypoxia-induced mitophagy in mammals, confirming the evolutionarily conserved function of upstream MAPK signaling pathway for regulating mitophagy from yeast to mammals [94]. Surprisingly, both Hog1 and p38 in yeast and mammalian cells failed to translocate to mitochondria and instead remained in the cytosol in response to mitophagy-inducing conditions [39,94]. These results raised the possibility that p38 may have cytosolic substrates that directly/indirectly phosphorylate unknown mitophagy regulators (e.g., Bcl-2-L-13; mammalian counterpart of yeast mitophagy receptor Atg32), which can allow recruitment of mitophagy machinery.

In addition to the conserved MAPK signaling pathway, the tumor suppressor p53, a multifunctional transcription factor, has been shown to regulate mitophagy. p53 is activated by a variety of cellular stresses, including genotoxic stress, oxidative stress, ribosomal stress, hypoxia, and starvation [95]. Nuclear p53 can positively and negatively regulate the expression of its target genes, resulting in several cellular responses by different stress signals. Interestingly, p53 functions as a negative regulator of the PINK1–PARKIN mitophagy pathway. Nuclear p53 represses the transcription of PINK1 [96], whereas cytosolic p53 can sequester and inhibit PARKIN translocation to mitochondria [97]. Thus, genetic and pharmacological inhibition of p53 can activate PARKIN-dependent mitophagy, preserving mitochondrial integrity and protecting against glucose tolerance, heart failure, and cardiac aging in mice [97,98].

In addition to p53, the redox-sensitive transcription factor Nrf2 (nuclear factor erythroid 2-related factor 2) is also involved in mitophagy regulation by binding to a putative antioxidant response element (ARE) of target genes [99]. Under basal conditions, Nrf2 is efficiently ubiquitinated by Keap1 (kelch-like ECH-associated protein 1)-Cul3 E3 ligase complex and constitutively degraded via the ubiquitin-proteasome pathway [99]. However, in response to stress (e.g., oxidative stress), Keap1 undergoes conformational changes in its cysteine residues, leading to its inactivation. Consequently, Nrf2 is stabilized and translocates into the nucleus to initiate the transcription of several cytoprotective genes, including PINK1 and p62 [100,101]. Thus, activation of the Nrf2-ARE signaling pathway can directly regulate mitophagy. Indeed, PMI (p62–SQSTM1-mediated mitophagy inducer), a synthetic compound, promotes p62-mediated mitophagy via Nrf2 stabilization in mammalian cells [102].

Besides the well-studied PINK1–PARKIN system, PARKIN-independent mechanisms have also been identified to promote mitophagy in mammals that rely on ubiquitin-independent mitophagy receptors. These receptors include NIX/BNIP3L, FUNDC1, AMBRA1, Prohibitin 2 (PHB2), MCL-1, cardiolipin (CL), ceramide, FKBP8, ATAD3B, and Bcl2-L-13 (mammalian homolog of Atg32) that directly bind to LC3 family proteins via LIR motifs and regulate selective mitochondrial clearance by sequestering mitochondria into autophagosomes [23,24,33,103,104,105,106,107,108,109,110]. For instance, Nix-dependent mitophagy regulates RBC maturation from erythroid precursor cells [23,24] and FUNDC1 mediates mitochondrial clearance upon hypoxia [103]. Note that these mitophagy receptors are often post-translationally modified for their efficient interaction with LC3 during mitophagy. In addition, a recent report showed that two mitochondrial matrix proteins, NIPSNAP1 and NIPSNAP2, can serve as autophagy-related receptors by accumulating on the mitochondrial surface in response to mitochondrial depolarization [111]. NIPSNAP1 and NIPSNAP2 function as “eat-me” signals for mitophagy by interacting with LC3/GABARAP proteins, autophagy receptors, NDP52, p62, NBR1, TAX1BP1, and autophagy adaptor ALFY, as well as with PARKIN [111]. NIPSNAP1 and NIPSNAP2 are new players of mitophagy that have a neuroprotective function in vivo. For example, NIPSNAP1-deficient larvae of zebrafish model organism show neurodegenerative phenotypes, including loss of tyrosine hydroxylase (Th1)-positive dopaminergic neurons, increased oxidative stress, increased cell death (apoptosis), and reduced locomotor activity as a consequence of impaired mitophagy [111]. Such phenotypes were also observed in the zebrafish model upon loss of PINK1 function [112], showing the relevance of NIPSNAP1 activity in the PINK1–PARKIN mitophagy pathway in the brain. However, it is unclear how NIPSNAP 1 and NIPSNAP2 are stabilized on the mitochondrial surface and whether NIPSNAP proteins can impact PARKIN-dependent ubiquitination of MOM substrates in multiple cell types under different mitochondrial stress conditions.

### 2.9. Ubiquitin-Dependent Mitophagy in Yeast

Ubiquitin-dependent mitophagy is well-studied in mammals, as discussed above. We recently identified 96 mitophagy regulators (86 positive and 10 negative) from a combined genetic and biochemical high-throughput screen [30]. The cytosolic Ubp3-Bre5 deubiquitinase complex is recruited to mitochondria upon mitophagy induction in respiring cells by rapamycin (Figure 4). The Ubp3-Bre5 complex inhibits mitophagy, while it promotes different types of selective autophagy, including bulk autophagy, the CVT pathway, and ribophagy [30,113]. Cells lacking Ubp3 or Bre5 show ∼1.5–2.0-fold higher rapamycin-induced mitophagy than the wild-type cells. However, substrates of the Ubp3-Bre5 complex have not yet been identified that can show a reciprocal response for mitophagy and other selective autophagy pathways. Thus, future studies will define the molecular mechanisms by which the ubiquitination and deubiquitination machinery can regulate mitophagy to allow yeast cells for adaptation under different growth conditions [114]. For instance, PARKIN was translocated to mitochondria upon oxidative stress or aging and extended the chronological life span (CLS) and oxidative stress resistance of the respiring yeast cells via mitophagy initiation [114].

### 2.10. Mitophagy in Neurodegenerative Diseases

It has been observed that loss-of-function mutations in PINK1 or PARKIN gene lead to defective mitophagy and the accumulation of dysfunctional mitochondria, contributing to autosomal recessive Parkinson’s disease (PD) [115,116]. Interestingly, these proteins also function in the same pathway to prevent mitochondrial damage in the *Drosophila* model, and defects in mitophagy result in reduced lifespan, apoptotic muscle and dopaminergic neuron degeneration, male sterility, fragmentation of mitochondrial cristae, and hypersensitivity to multiple stresses including oxidative stress and endoplasmic reticulum (ER) stress [117,118,119].

Ser65-phosphorylated ubiquitin (pSer65-Ub)-positive mitophagy granules were found to be accumulated in the human brain during aging and in PD patients with Lewy body disease [74]. Lewy bodies (LBs) that are α-synuclein-rich cytosolic inclusions serve as a pathological hallmark of PD and many other neurodegenerative diseases [120]. LB-like inclusions often trap fragmented membranes, vesicular structures, and organelles (mitochondria, autophagosomes, and lysosomes), and impair pathways of protein degradation or mitochondrial homeostasis [120]. In addition, pSer65-Ub were increased in brain tissue of mice model lacking PARKIN and bearing mutations in POLG (the catalytic subunit of the mitochondrial DNA polymerase gamma), which leads to mitochondrial dysfunction as a result of mtDNA mutations, premature aging, and defective respiratory chain assembly [121]. Thus, in addition to α-synuclein, pSer65-Ub serves as promising biochemical and imaging biomarkers of PD pathology. A low cellular abundance of pSer65-Ub from cerebrospinal fluid or blood might be detected by mass spectrometry-based phosphoproteomics to identify PD patients with defective mitophagy [122].

Mitophagy defects are not limited to the pathogenesis of Parkinson’s disease (PD), but are also involved in the pathology of other neurodegenerative diseases, including Alzheimer’s disease (AD), Huntington’s disease (HD), amyotrophic lateral sclerosis (ALS), and multiple sclerosis (MS), which are characterized by progressive degeneration of neurons, resulting in cognitive impairment and memory loss [123]. The pathological hallmarks of AD are the intracellular accumulation of amyloid-*β* (A*β*) plaques/aggregates and neurofibrillary tangles (NFTs), composed of hyperphosphorylated Tau (p-Tau) [123,124,125,126]. Both Aβ plaques and NFTs lead to mitochondrial impairment by different mechanisms, such as perturbation of oxidative phosphorylation (OXPHOS), alteration of mitochondrial dynamics, and the loss of mitochondrial proteostasis [123,124,125,126]. Thus, dysfunctional mitochondria are progressively co-accumulated with Aβ peptides in neurons of AD patients [126]. A recent study showed that mitophagy induction rescues A*β* and tau pathology in transgenic *C. elegans* and mice models of AD [127]. Thus, PINK1–PARKIN-mediated mitophagy plays a protective role in eliminating neurotoxic Aβ species and defective mitochondria at both neuronal and organismal levels.

Mitochondrial dysfunction is also associated with HD pathogenesis caused by aberrant expansion of CAG repeat in the coding region of the huntingtin (*HTT*) gene, resulting in expanded polyglutamine (polyQ) aggregation and neuronal death [123]. It is shown that basal mitophagy is markedly reduced in mice models of HD (i.e., HTT expressing mice), suggesting defective mitophagy is one of the causes of HD progression [128]. Interestingly, HTT-induced neurodegeneration was partially rescued upon PINK1 overexpression in fly and mice HD models [129]. Therefore, there is an increasing interest in stimulating mitophagy (PINK1–PARKIN-dependent and -independent) to treat HD and other neurodegenerative diseases.

Amyotrophic lateral sclerosis (ALS) and frontotemporal dementia (FTD) share a common mechanism for mitochondrial toxicity and neurodegeneration caused by protein misfolding and aggregation of mutant superoxide dismutase 1 (SOD1), TAR DNA-binding protein 43 (TDP-43), and RNA-binding protein FUS [123,130]. A relatively low PARKIN expression was observed in ALS/FTD mice model expressing mutant human TDP-43, suggesting an increased vulnerability to mitochondrial dysfunction and defective PINK1–PARKIN pathway linked to ALS/FTD pathogenesis [131]. In addition, through a poorly defined mechanism, mitophagy was suggested to regulate the pathogenesis of multiple sclerosis (MS), the most common neurodegenerative disease in young adults [132].

## 3. Conclusions and Perspectives

Mitophagy is a multistep process that maintains cellular and organismal fitness during aging or intracellular/environmental stress conditions. Moreover, mitophagy coordinates with mitochondrial biogenesis and dynamics to maintain mitochondrial homeostasis [133]. For example, mitophagy impairment compromises stress resistance, longevity, and mitochondrial function in *C. elegans* animal model [133]. Indeed, mitophagy-deficient animals show phenotypes such as elevated mitochondrial reactive oxygen species (ROS), increased mitochondrial DNA mutations, decreased ATP levels, mitochondrial membrane depolarization, and elevated cytoplasmic Ca^2+^ concentration [133]. Furthermore, neuronal mitophagy declines during human pathologies and ageing, leading to an accumulation of defective organelles [128,133]. Understanding the molecular mechanisms underlying mitophagy pathways in yeast and metazoans will have potential value to modulate mitophagy for therapeutic intervention in neurodegenerative diseases. Recently, new small-molecule compounds were identified that could amplify the catalytic activity of PINK1 and PARKIN (WT and PD-related mutant), and thus offer an effective therapeutic strategy to manipulate and rescue the efficient clearance of defective organelles via mitophagy in PD patients [134,135]. However, kinetin triphosphate (KTP), an ATP neo-substrate, can only pharmacologically boost or restore PINK1 activity upon mitochondrial depolarization by CCCP, which may limit its application under the pathophysiological setting [135]. Further studies are required to validate the drug efficacy of these compounds in rodent genetic PD models. Furthermore, PINK1 and PARKIN activity may be modulated by screening and identifying potential synthetic or natural small-molecule inhibitors against specific DUBs (USP15, USP30, and USP35) that antagonize the PINK1–PARKIN pathway. Alternatively, the pharmacological activation of USP8 by small-molecule-compounds may increase mitophagy in PD patients with decreased PINK1 or PARKIN activity.

## Figures and Tables

**Figure 1 ijms-22-04363-f001:**
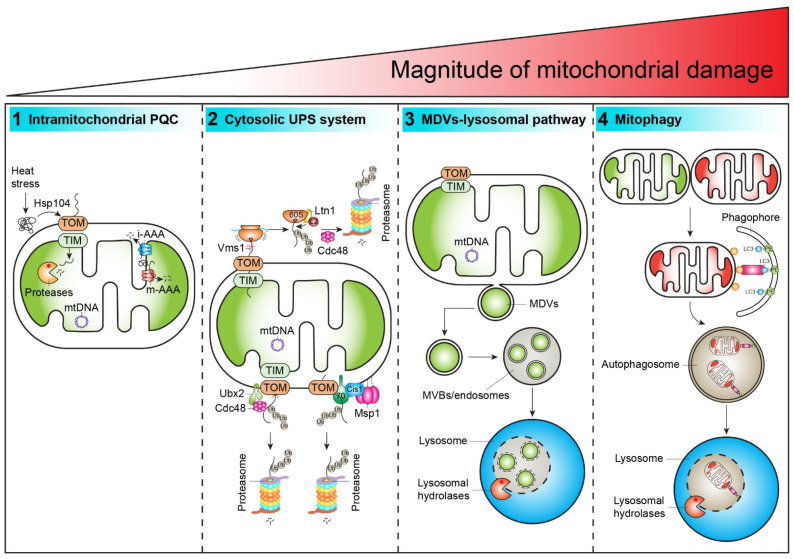
Mitochondrial quality control pathways. In response to moderate mitochondrial damage, misfolded or damaged proteins are initially degraded by the mitochondrial protease–chaperone network (**1.**). The inner mitochondrial membrane-localized AAA^+^ metalloproteases (i-AAA and m-AAA) and the mitochondrial matrix localized ATP-dependent Lon (Pim1 in yeast) protease are main players removing misfolded/damaged proteins. Heat stress-induced aggregation-prone cytosolic proteins are partly imported into mitochondria for degradation in a Hsp104-dependent disaggregation manner. Stalled mitochondrial precursor proteins are degraded via distinct pathways following ubiquitination and proteasomal degradation (**2.**). These cytosolic ubiquitin-proteasome system (UPS) pathways prevent an accumulation of stalled precursors on the organelle’s surface, safeguarding the translocation channel from being clogged, and thus restore full protein import capacity. Upon mild/local mitochondrial oxidative damage, mitochondrial-derived vesicles (MDVs) are generated that are degraded in the lysosome (**3.**). In response to severe mitochondrial damage or when other quality control pathways fail, the entire organelle is selectively removed via mitophagy at the cellular level (**4.**). All levels of mitochondrial quality control presumably occur in parallel, yet the exact interplay and regulation is a matter of current research.

**Figure 2 ijms-22-04363-f002:**
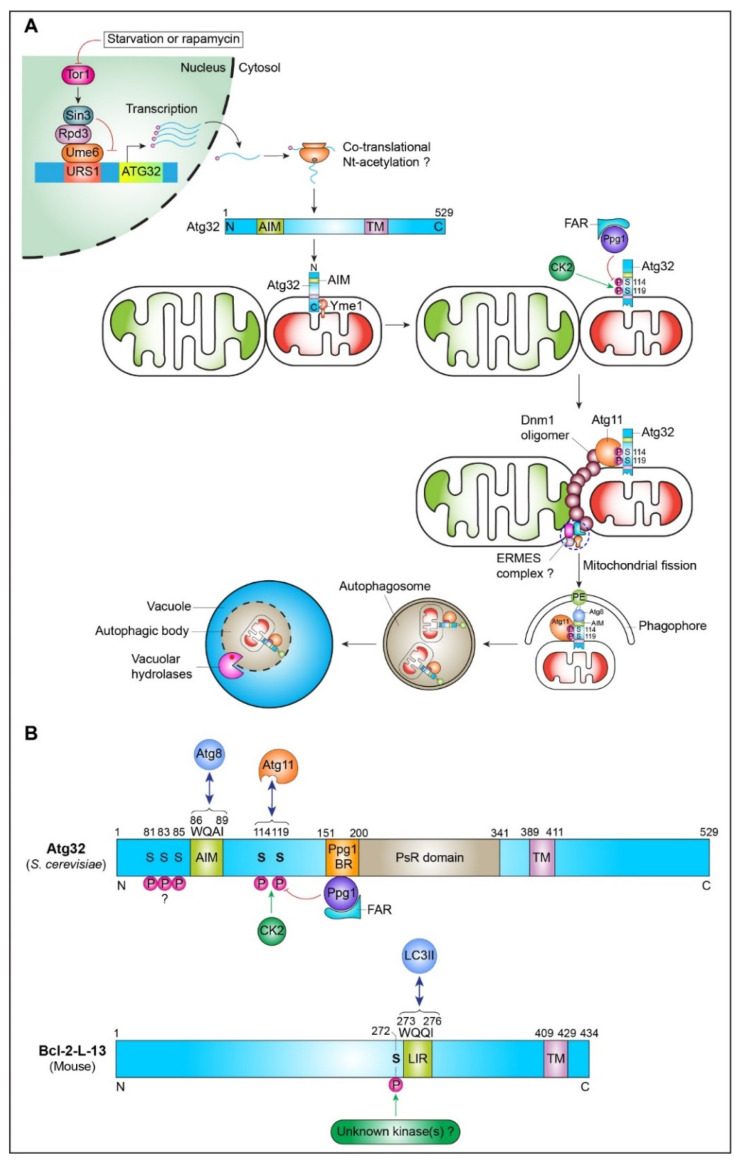
Molecular mechanism of Atg32-mediated mitophagy. (**A**) Regulation of Atg32 activity. Upon starvation or treatment with rapamycin, Tor1 (target of rapamycin 1), an evolutionarily conserved Ser/Thr kinase, is inactivated, releasing Sin3–Rpd3–Ume6 transcription repressor complex from the ATG32 promoter, and thus allowing a strong transcriptional induction of ATG32. Translated Atg32 is translocated to mitochondria and proteolytically processed by the Yme1 i-AAA protease at its C-terminus. Subsequently, Atg32 is phosphorylated by the Casein kinase 2 (CK2) at two residues, Ser114 and Ser119, that allow interaction with the scaffold protein Atg11 and Atg8 at the PAS. Atg11 may also recruit Dnm1 and the ERMES complex components for mitochondrial fission before mitochondria are sequestered into double-membrane autophagosomes. Atg32 phosphorylation is reversed by a phosphatase Ppg1 and the Far complex. Note that Yme1-dependent processing of Atg32 was observed when mitophagy was initiated by nitrogen starvation. (**B**) Schematic diagram of functional domains of Atg32 (*S. cerevisiae*) and Bcl-2-L-13 (mouse). Atg8-interacting motif/LC3-interacting region (AIM/LIR); Ppg1 binding region (Ppg1 BR); pseudo-receiver (PsR) domain; transmembrane domain (TM); S (Ser). The potential phosphorylation sites (S81/83/85) in Atg32 are shown upstream of the AIM motif. The kinase (s) that phosphorylates Bcl-2-L-13 at Ser^272^ upstream of the functional LIR motif is still unknown. The protein size is shown by the number of amino acid residues.

**Figure 3 ijms-22-04363-f003:**
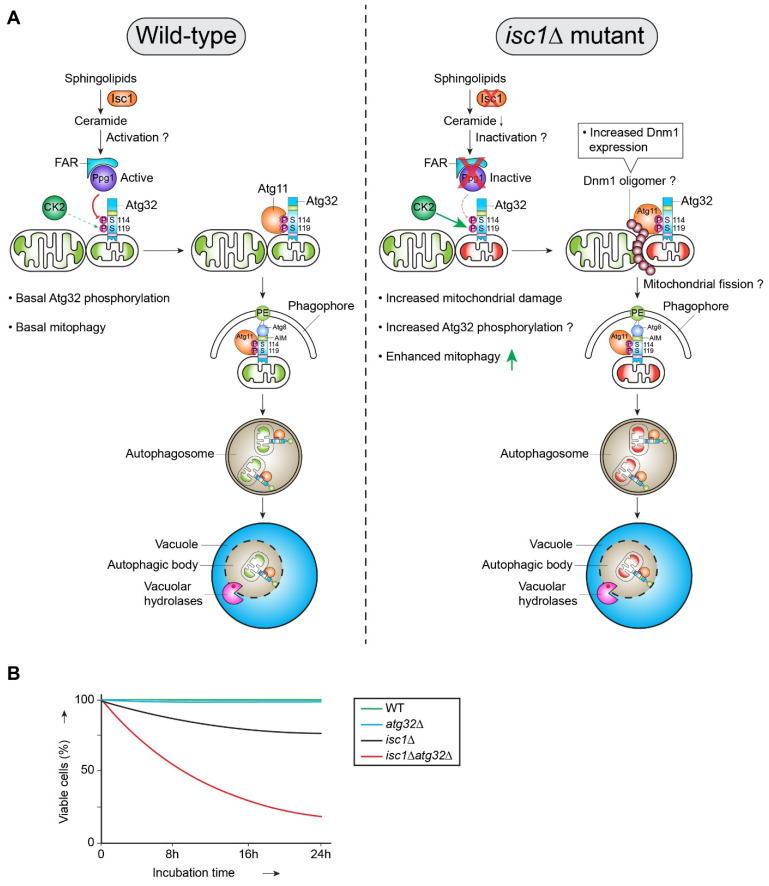
Interplay of ceramide synthesis, mitochondrial damage, and mitophagy in baker’s yeast. (**A**)**.** Regulation of Atg32-dependent mitophagy in the wild type (WT) and *isc1*∆ mutant strains. *S. cerevisiae* WT cells, when grown over time on non-fermentable carbon sources, maintain a basal level of mitophagy limiting excessive mitophagy. This apparently depends on the recruitment of Isc1 to mitochondria that catalyzes ceramide synthesis from sphingolipids. Mitochondrial ceramides, in turn, may activate mitochondrial localized protein phosphatase Ppg1, thus counteracting CK2-mediated phosphorylation of Atg32 and limiting excessive mitophagy. The *isc1*Δ cells show mitochondrial damage and fragmentation linked to increased protein expression of Dnm1. The *isc1*∆ mutant cells show enhanced Atg32-mediated mitophagy promoting cellular viability. (**B**) Isc1 and Atg32-dependent cellular viability. The mitophagy-deficient *atg32*Δ strain does not show any defect in the cell growth and viability. The ceramide-deficient *isc1*Δ strain can still maintain cell viability (∼80% of the wild-type level) by enhancing mitophagy, which acts as a protective mechanism here. However, additional deletion of *ATG32* in *isc1*∆ cells (i.e., *isc1*Δ*atg32*Δ double mutant) drastically reduces cell survival as Isc1-dependent enhancement of mitophagy cannot operate in the absence of Atg32.

**Figure 4 ijms-22-04363-f004:**
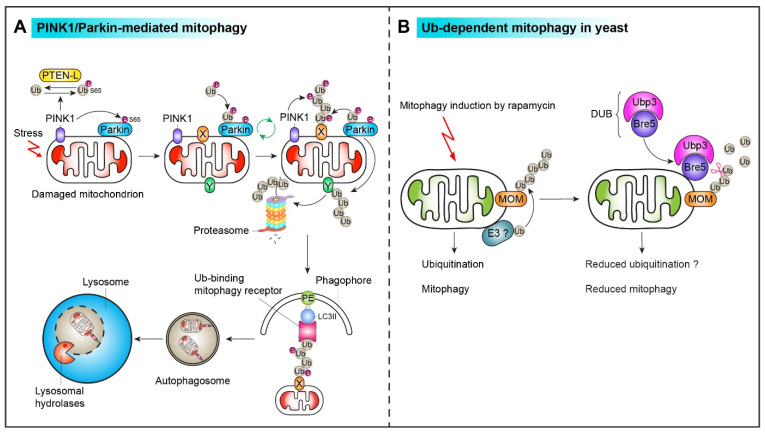
Ubiquitin-dependent mitophagy in mammals and yeast. (**A**) The PINK1–PARKIN pathway. Upon mitochondrial damage, PINK1 kinase is stabilized on the mitochondrial surface and recruits the E3 ubiquitin ligase PARKIN. PINK1 phosphorylates both ubiquitin (Ub) and PARKIN at the residue Ser65 and generates pSer65-Ub and pSer65-PARKIN, respectively. Mitochondrial localization of PARKIN and its ubiquitin ligase activity is enhanced drastically (∼4400-fold) when pSer65-Ub binds to pSer65-PARKIN. This can create a feedforward loop by providing additional ubiquitin molecules for PINK1 phosphorylation. Some PARKIN substrates (labeled as “Y”) are degraded by the ubiquitin-proteasome system (UPS) during mitophagy. Mitochondria decorated with polyubiquitin/phospho-ubiquitin chains (shown on “X”) are recognized by ubiquitin-binding mitophagy receptors and sequestered by autophagosomes for lysosomal degradation. (**B**) Ub-dependent mitophagy in yeast. Upon mitophagy induction in respiring cells with treatment with rapamycin, mitochondrial outer membrane (MOM) proteins can be ubiquitinated by unknown E3 ubiquitin ligase(s). The autophagy machinery recognizes ubiquitinated mitochondria for subsequent vacuolar degradation. The cytosolic Ubp3-Bre5 deubiquitinase complex can inhibit mitophagy when it is recruited to mitochondria presumably by removing ubiquitin moieties from MOM proteins.

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
