# Peer review of "Common Principles and Specific Mechanisms of Mitophagy from Yeast to Humans"

_ijms, 2021, doi:10.3390/ijms22094363_

Round 1

Reviewer 1 Report

The manuscript of Kumar and Reichert is dedicated specific mechanisms of mitophagy regulation of yeast and mammals. Mitochondrial quality control is a key process for maintenance of healthy cells and organisms. Mitophagy is the only known cellular pathway, which damaged mitochondria are completely eliminated. In this review authors discuses mechanisms of mammals and yeast mitophagy and its regulation in a ubiquitin-dependent and -independent manner. The article is well written, well structured. I would like to note the high quality of the figures, which describe in detail the schemes of mitophagy regulation.

Despite the exhaustive content of the manuscript, there are several uncovered issues that may be important and interesting.

  1. There is insufficient data on p62 (SQSTM1) and its involvement in mitophagy. P62 is an important regulatory molecule that functions as a selective autophagy receptor for the degradation of ubiquitinated substrates. p62 and PINK1/PARKIN plays an important role in the selective elimination of damaged mitochondria. Also, there is p62-mediated mitophagy and can also be carried out in the PINK1/PARKIN-independent pathway (PMID: 25013806).
  2. The authors mention the role of mitogen-activated protein kinase (MAPK) signaling cascades in the regulation of mitophagy. There is no mention of the connection between mitophagy and other signaling pathways. For example, p53 play an important role in mitophagy controls (PMID: 29033320) and authors should discuss it.
  3. Also, it is known that Nrf2/ARE signal pathway direct regulated expression of some key protein involved in mitophagy, such as PINK1 (PMID: 25013806) and p62 (PMID: 25013806).
  4. There is chapter, that named “Mitophagy in neurodegenerative disease”. However, the authors describe only the effect of mitophagy on Parkinson's disease, although there are mentions that mitophagy defects are involved in the pathogenesis of Alzheimer's disease (PMID: 25013806), amyotrophic lateral sclerosis PMID: (33450997), multiple sclerosis (PMID: 31248423) and other neurodegenerative diseases. 

Reviewer 2 Report

This review is well written, and easy to understand the contents. The authors describe the mechanisms of mitophagy from yeast to human in a balanced manner. The followings are my comments.  

  1. In the legend to figure 1, the authors should describe what i-AAA and m-AAA stand for respectively.
  2. The authors need to explain what the abbreviation stands for the first time it is used in their manuscript. For example, OMM and IMM at the line 107, and MOM at the line 439.
  3. Is the word “OPA” correctly OPA1 at the line 115?
  4. It might be better to comment on the interaction between OPTN-ATG9A.

Round 2

Reviewer 1 Report

The authors have significantly revised the manuscript. The authors have added new information that expands the review. I believe that this version of the article can be accepted.

Reviewer 2 Report

The revised manuscript by Kumar and Reichert is substantially improved over the original manuscript. I am satisfied with all but one their response.

 1. The abbreviation “MOM” appears at the line 441 for the first time. Please check the line 443.